# Modeling of the Stress–Strain of the Suspensions of the Stators of High-Power Turbogenerators

**Oleksii Tretiak** [1,*], **Dmitriy Kritskiy** [2,*], **Igor Kobzar** [3,*], **Victoria Sokolova** [4,5,*], **Mariia Arefieva** [5,6,*], **Iryna Tretiak** [7,*], **Hromenko Denys** [1,*] and **Viacheslav Nazarenko** [1,*]

1 Faculty of Aircraft Engineering, Department of Aerohydrodynamics, National Aerospace University «Kharkiv Aviation Institute», 61070 Kharkiv, Ukraine
2 Department of Information Technology Design, National Aerospace University «Kharkiv Aviation Institute», 61070 Kharkiv, Ukraine
3 Special Design Office for Turbogenerators and Hydrogenerators, Joint Stock Company "Ukrainian Energy Machines", 61037 Kharkiv, Ukraine
4 NNI «Institute of Public Administration», V. N. Karazin Kharkiv National University 61070 Kharkiv, Ukraine
5 Department of Aerohydrodynamics, National Aerospace University «Kharkiv Aviation Institute», 61070 Kharkiv, Ukraine
6 Kharkiv Lyceum «IT Step School Kharkiv», 61010 Kharkiv, Ukraine
7 Aerospace Thermal Engineering Department, National Aerospace University «Kharkiv Aviation Institute», 61070 Kharkiv, Ukraine
* Correspondence: alex3tretjak@ukr.net (O.T.); d.krickiy@khai.edu (D.K.); ivkobzar@ukr.net (I.K.); sokolova.v.mk@gmail.com (V.S.); marii.arefieva@gmail.com (M.A.); irina.ii3t@gmail.com (I.T.); dvgromenko@gmail.com (H.D.); my_registrator@ukr.net (V.N.)

**Abstract:** In the submitted scientific work, the existing types of stator fastening design of turbogenerators and the main causes of the stressed state of the stator suspensions are considered. A detailed calculation of the complex stressed state of the turbogenerator stator suspension was carried out for a number of electrical sheet steels, taking into consideration the unevenness of the heat distribution along the horizontal axis of the unit. It is proposed that the calculation of the mechanical stress is carried out by means of the mechanical and thermal calculation, coordinated with the electrical one. The possibility of replacing steel 38X2H2BA with steel 34CrNiMo6 and 40NiCrMo7 is indicated, subject to compliance with GOST 8479-70 for the same strength group.

**Keywords:** turbogenerator; spring suspension; limiting conditions; active steel; the stator casing





## 1. Introduction

In recent decades, there has been a tendency not only to increase the efficiency of electric machines (EM), but also to decrease their mass-dimensional indices per unit of power. As a rule, this is carried out by the optimization of calculations and the use of three-dimensional modeling of physical processes (electromagnetic, mechanical, temperature, ventilation, etc.), as well as the emergence of new materials with improved parameters, which allows reduction of the weight of turbogenerator units.

Today, in Ukraine, almost 74% of electrical energy is produced at thermal and nuclear power plants, where turbogenerators are in operation. The service lives of most turbogenerators have already expired, and others are on the verge of expiration; this is due to long-term insufficient financing of the energy industry. At the same time, EM operating modes are complicated by uneven loads in the electrical network, which cause both generator overload (transition to emergency modes of operation, due to malfunction of generators at stations or an increase in the amount of consumed energy), and their shutdown (decrease in the amount of consumed energy). The solution to this problem is the partial modernization of already existing units with an increase in their capacity

and the parallel, step-by-step replacement of the rest of the outdated machines with more powerful and lighter ones.

The development of a single methodology for calculating the stress–strain state (STS) for high-power turbogenerator units, based on a combination of analytical and three-dimensional calculations that allows increasing the accuracy of the calculation problem, is of great scientific and practical interest.

Let us consider the design of a two-pole turbogenerator manufactured by the EM WEG Group (USA) (see Figure 1) [1]. Application of the designs by putting into practice additional flexible elements, as a rule, is the most suitable method for turbogenerators rated over 100 MW.

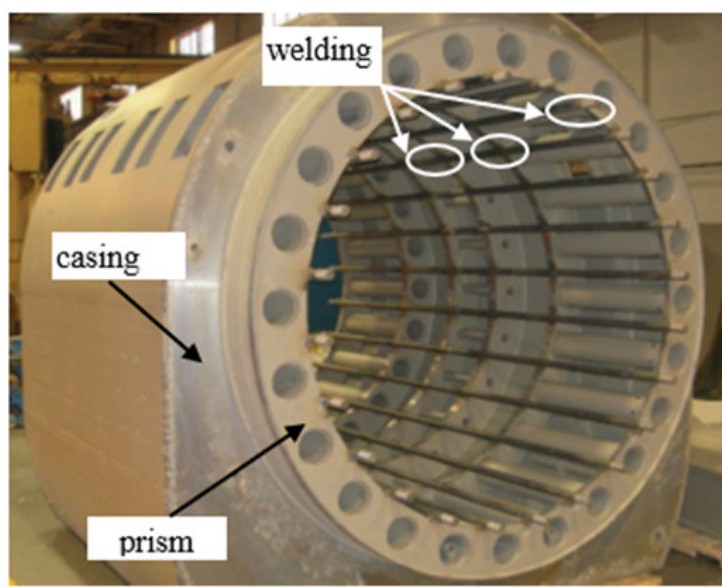

**Figure 1.** Stator Casing of EM WEG Group.

The core fastening units include a system of prisms, which, with the help of pressing down flanges, keep the assembled core in a monolithic state. However, there are no springs in these suspensions that dampen the vibrational components of the harmonics of electric forces caused by gravity.

Classic designs of turbogenerators include two types of suspension, namely internal and external. One of the most striking and reliable representatives of turbogenerators with internal suspension are Turbogenerators rated 200 MW and 300 MW of the TGV series produced by the JSC "Ukrainian Energy Machines". The design of the stator housing of an electric machine has been submitted in the scientific paper [2]. In Refs. [3,4] the results of calculations of thermal fields in the end parts of the turbogenerator are presented. However, as operating experience shows, active cooling inside the rods and the high speed of the flowing refrigerant solve the problem of lowering temperatures in the end parts and their fastenings for the winding rods. Based on operating experience, the numerical solution for the frontal parts does not give an answer regarding the state of the grooves. Figure 2 shows a serial sample and a longitudinal section.

Vibration is one of the important factors leading to the damage and destruction of turbogenerator housing parts. In Ref. [5], the effect of vibration on turbogenerator assemblies is considered in detail. In turbogenerator design, the suspension (plate springs) takes the load caused by vibration forces and must ensure the safety of the structure when the unit enters the short-circuit mode. Other methods require optimization of the design to increase power while maintaining weight and dimensions [1,2,6]. When solving these problems, it is necessary to pay attention to the level of vibration when carrying out tests [7]. To ensure reliable operation, it is necessary to calculate the design of the suspension, which reduces vibration while maintaining the basic power.

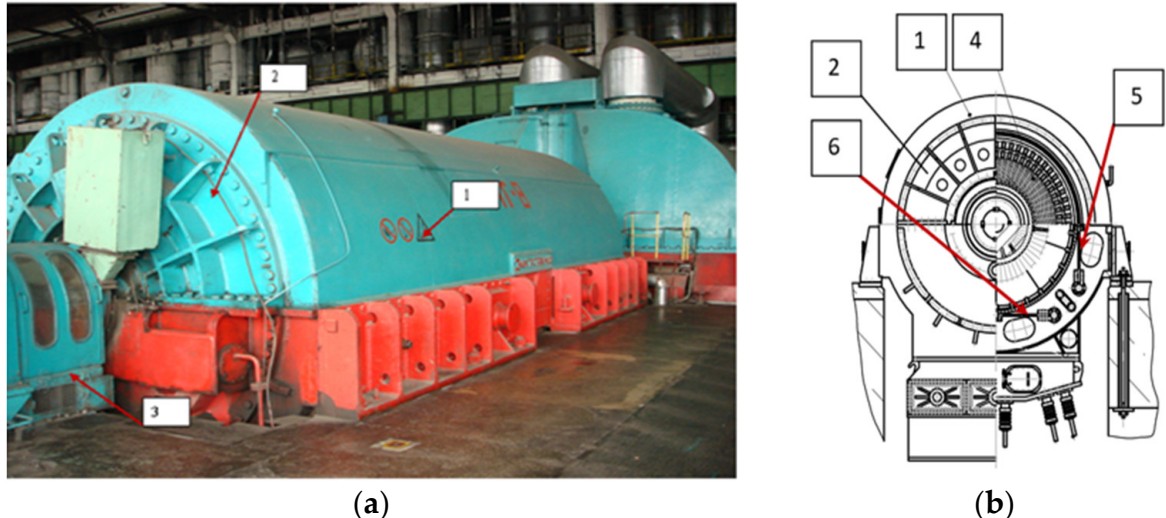

(**a**)                                                    (**b**)

**Figure 2.** Turbogenerator type TGV-300: (**a**) at the base of operations, TPP; (**b**) cross section: 1—stator casing; 2—external shield; 3—brush-holders device; 4—stator (active steel and bars); 5—vertical row of springs; 6—horizontal row of springs.

## 2. Turbogenerator Suspension Design by JSC "Ukrainian Energy Machines"

In the given scientific paper, the strength of the structures of the internal suspensions of the stators of hydrogen-cooled turbogenerators rated 200 MW, 250 MW, 300 MW, and 325 MW, produced by JSC "Ukrainian Energy Machines" is studied. All the researched machines have a similar design of internal suspensions; the difference is in the number and geometric parameters of the suspension used.

In Figure 3, the general view of the turbogenerator is submitted.

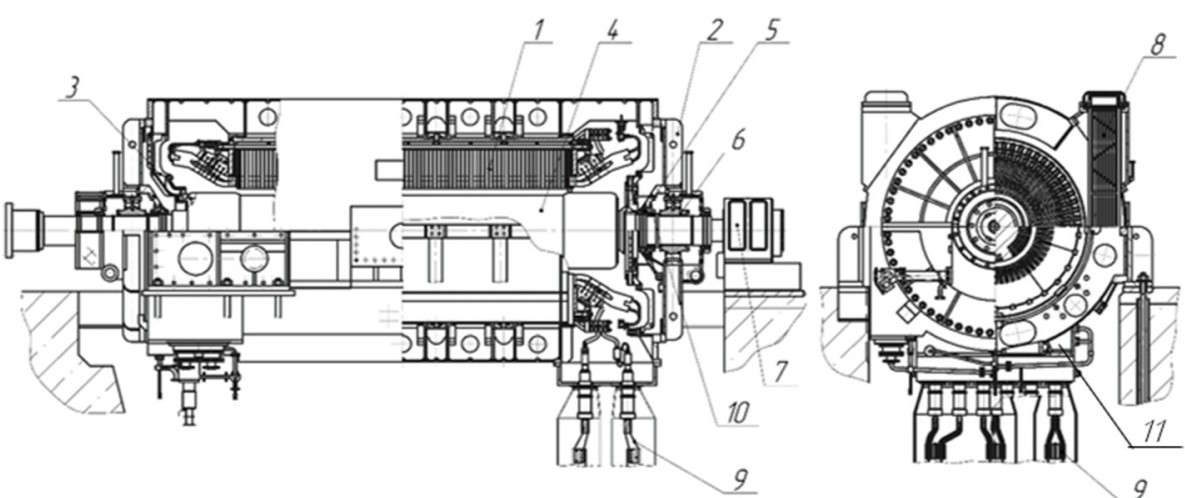

**Figure 3.** Turbogenerator TGV-200 design includes the following, namely: 1—stator; 2—external shield; 3—internal fairing; 4—rotor; 5—rotor shaft sealing; 6—bearing; 7—brush-holder device; 8—gas-cooler; 9—end terminals; 10—oil trap; 11—stator suspension.

An analysis of the stress–strain state of the suspension unit, which includes a spring, a support plate, a strap, and a system of pins and bolted connections, was carried out (see Figure 4).

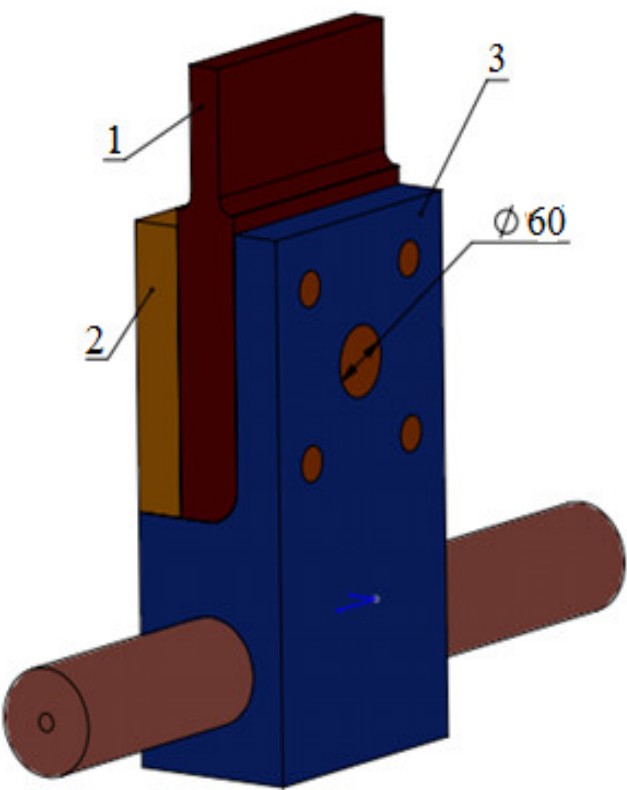

**Figure 4.** Three-dimensional image of the assembly of the spring to the frame: 1—spring; 2—support plate; 3—system of pins.

The strength of the suspension unit was studied at the moment of a two-phase short-circuit, which corresponds to the maximum load on the suspension system. The occurrence of a short-circuit is characterized by the emergence of a moment of short-circuit of the MRC, which leads to compressive/stretching forces of the RKZ acting on the springs. The magnitude of these forces is determined for each machine by classical analytical methods of calculating the suspension of the stator core during a two-phase short-circuit, in accordance with the technical conditions for Turbogenerators TGV-200, TGV-300 series, manufactured by JSC "Ukrainian Energy Machines". The calculation is carried out for the case of static loading of the suspension system by the compression/tension force of the RKZ, while a dynamic factor is chosen equal to 2.

The rated torque acting on the stator of the generator is determined by the following formula:

$$M_H = 9560\frac{N}{n} \, , \tag{1}$$

where $N$ is the power of the generator and $n$ is the rotational speed of the generator per minute (rpm).

Forces on the vertical spring at rated mode are calculated as per the following formula:

$$P_H = \frac{Gg}{z_B} + \frac{M_H \times cos\varphi}{z} \times \frac{1}{R} \, , \tag{2}$$

where $G$ is the stator mass, $z_B$ is quantity of vertical springs, $cos\varphi$ is the power factor, $z$ is the general quantity of the springs, and $R$ is the radius of the spring arrangement.

At short-circuit mode, this force is determined as:

$$P_{K3} = \frac{Gg}{z_B} + P_{max} \, , \tag{3}$$

where $P_{max}$ is force, which acts on the spring at a sudden two-phase short-circuit on the terminals.

In addition, the influence of temperature loads on the suspension is taken into account, which vary along the length of the stator core and are determined to take into consideration the internal heating of the stator with the solution of the gas-dynamic problem. According to the experience of operating turbogenerators rated 325 MW with hydrogen cooling, the temperature difference between the core and the stator casing can be 60 °C which is confirmed by calculation data. According to the calculation data for a turbogenerator rated 250 MW with hydrogen–water cooling, the temperature of the "active steel" of the stator on the side of the slip rings is 36 °C, in the middle of the machine is 39 °C, and on the side of the turbine is 41 °C. The gas temperature in the radial channel is in the range of 25 °C to 47 °C. The hottest point of the gas on the stator is 47 °C and is located in the second compartment in the area of the backrest. The gas temperature increases from the tooth to the back by ~10 °C. The turbogenerator cooling system is shown in Figure 5.

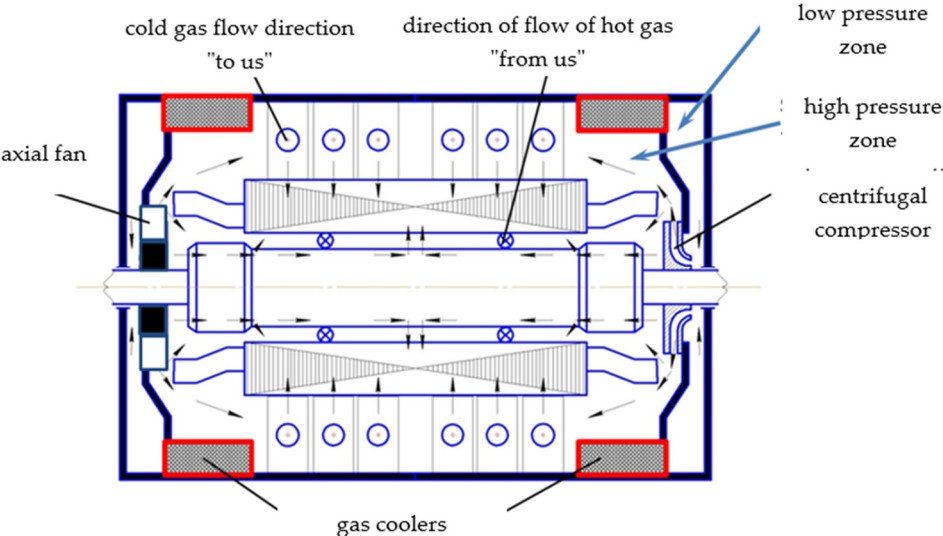

**Figure 5.** Turbogenerator cooling system.

The longitudinal ventilation channels are shown in Figure 6. The springs are inside the stator.

Therefore, for each row of springs, it is necessary to determine the mechanical stresses, taking into account the change in their thermal state. Determination of the temperature field in parts of the suspension unit is performed by solving the unsolved thermal problem, applying boundary conditions of the first kind.

Thus, the calculation of the suspension is carried out for the axial tension/compression forces acting on the spring at short-circuit and are determined according to the classical engineering methods used for the calculation of the suspension of turbogenerators. With that, the temperature loads on the suspension assembly unit mentioned above are taken into account. This allows for a more accurate description of the real stress–strain state (STS) in the suspension assembly unit.

The general view of the suspension of the considered turbogenerators is shown above in Figure 4. The suspension consists of vertical and horizontal flat springs, one end of which is fixed to the stator casing, and the other is fastened to the frame. The number of stator spring suspensions for generators, even those of the same power, may vary. At the same time, depending on the power of the generator, the geometric parameters of the suspensions and the load acting on them change. Further, a study of the strength of the suspension unit for generators rated 200 MW, 250 MW, and 325 MW is performed.

The purpose of the calculation is to define the stresses in the suspension spring and the details of its fastening to the stator casing and to the frame in case of a short-circuit in

the stator winding, taking into account the unevenness of temperature loads and possible assembly inaccuracies.

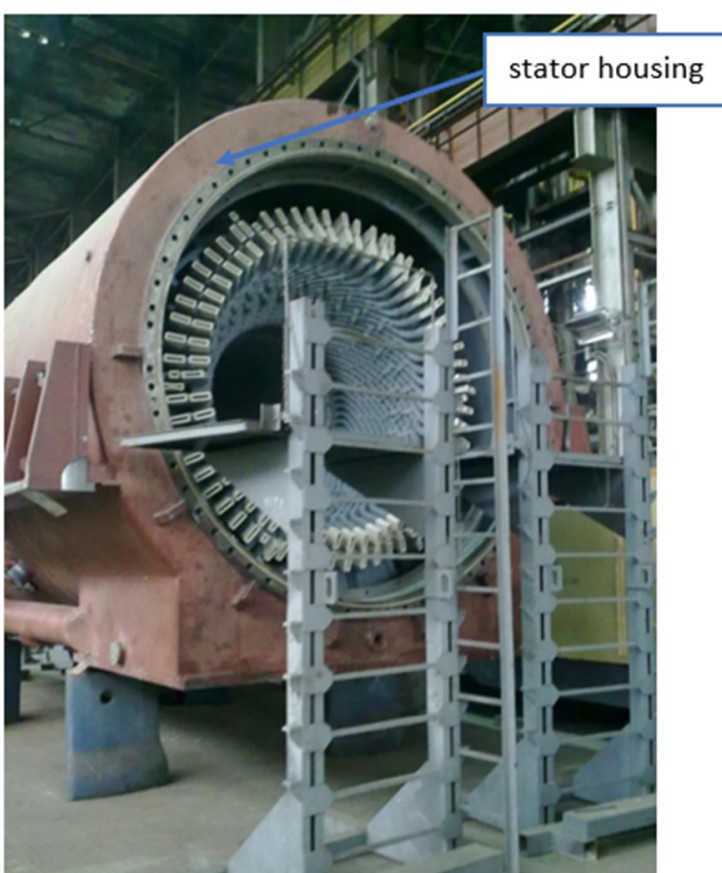

**Figure 6.** The Stator of Turbogenerator TGV-300-2U3.

### 3. Study of the Strength of the Stator Suspension of Generators Rated 325 MW and 250 MW

The main geometric and physical characteristics of the suspension of the turbogenerator rated 325 MW, 3000 rpm are as follows:

- weight of the stator core with the winding                    G = 185,000 kg;
- quantity of springs                                                       Z = 20 pcs.;
- thickness of a spring                                                    h = 1.8 cm;
- widths of a spring                                                        b = 20 cm;
- calculation length of a spring                                      l = 65 cm;
- cross-sectional area of a spring                                  F = b × h = 20 × 1.8 = 36 cm$^2$;
- the distance between the eye bolts of a spring            L = 85 cm;
- the radius of springs arrangement                              R = 147.4 cm.

The amplitude value of the moment during a short-circuit is equal to Msh.c. (MKZ) = 2.62 × 108 kg cm, and the maximum force on one spring from the torque during a short-circuit is P2 = 89,000 kg.

The stator core of the generator rated 325 MW is attached to the housing with 10 horizontal and 10 vertical springs. Fastening with the help of 16 springs is also used. Figure 7 shows the fastening unit for attaching the spring to the frame.

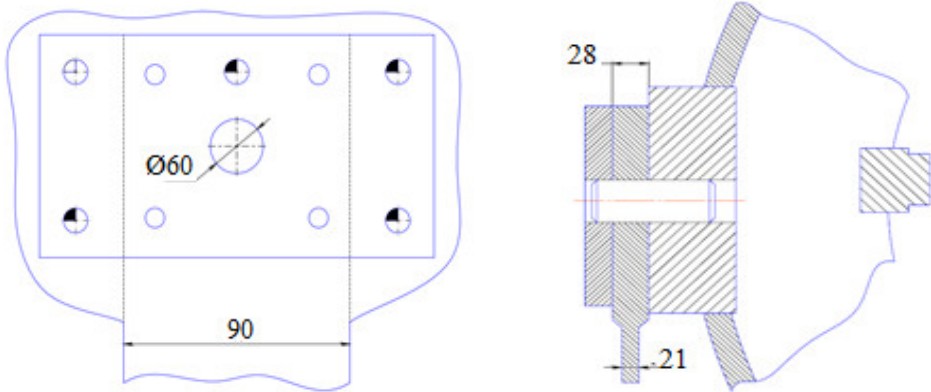

**Figure 7.** Drawing of the fastening unit for attaching the spring to the frame.

Similar images of the method of the fastening of the spring to the stator casing are shown in Figures 8 and 9.

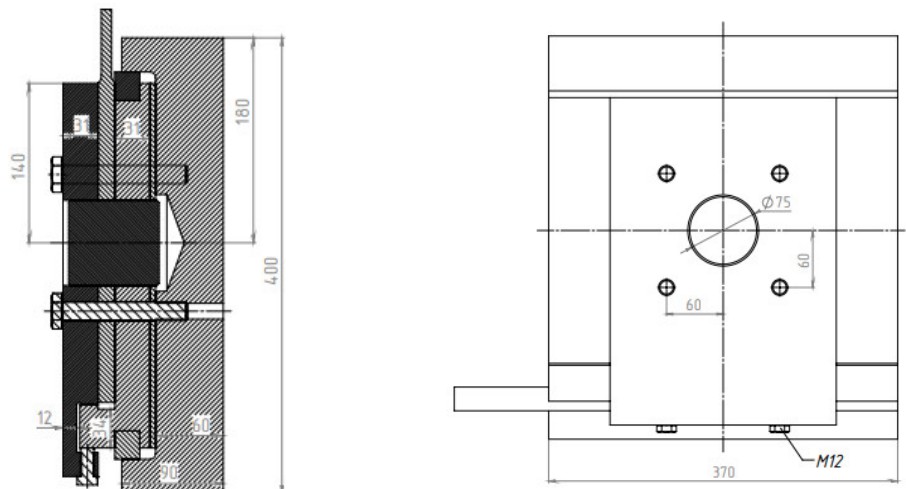

**Figure 8.** Drawing of the fastening unit of a spring to the stator casing.

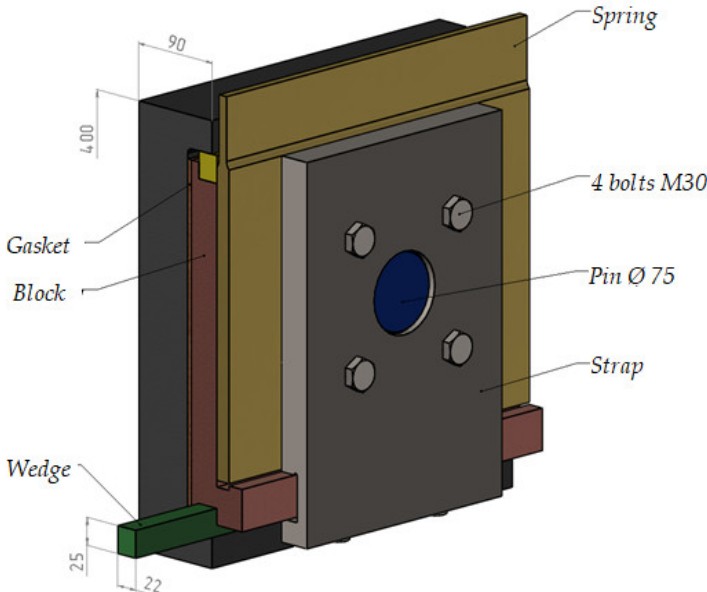

**Figure 9.** Three-dimensional image of the assembly of the spring attachment to the stator casing.

A plate with mounting holes is welded to the frame, to which a spring is attached and fixed with a strap. In the generator rated 250 MW and 320 MW, the spring is attached to the frame with a conical pin with a diameter of 60 mm, 5 conical pins of 30 mm, and 4 bolts, M30.

The spring is attached to the stator casing with a pin and a strap. In the generator rated 250 MW, the spring is attached for support with a conical pin with a diameter of 60 mm and three bolts, M30. A sleeve welded to the rings of the stator casing is installed on the cylindrical ends of the supports with tension. The conical pin is held on one end by a support, and on the other by a cheek, which is attached to the support with two bolts, M36, working in tension. Thus, the pin has two planes of cutting. On a generator rated 320 MW, fastening is carried out with one pin with a diameter of 60 mm and 4 bolts, M36.

The material of the suspension and pins is alloyed steel.

In Figure 10, a calculation diagram with the main loads acting on the support elements of the suspension and the temperature of its elements is shown, where the numbers 1, 2, and 3 indicate the support pins, and 4 indicates the main body of the spring, the directions of forces are indicated by arrows. T1, T2, T3, and T4 are the calculated temperatures of the suspension elements.

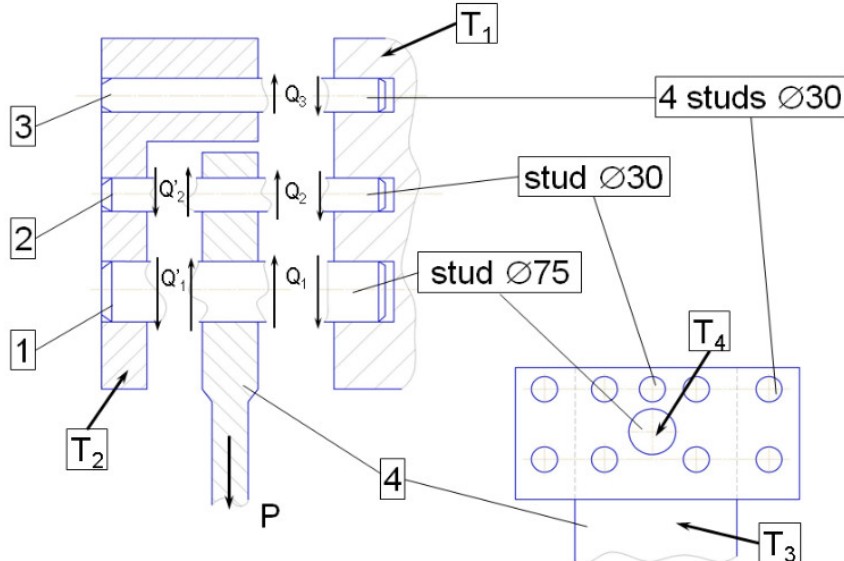

**Figure 10.** Calculation Diagram of the Suspension: 1, 2, 3—support pins; 4—spring.

One of the defining moments in solving MSE problems is the choice of a finite element. Two types of tetrahedrons (Figure 11) are used as basic finite elements with different approximations of movements inside the element. The first tetrahedron has units at the tops (Figure 11a) and is based on a linear approximation of movements inside the element, and the second is an oblique tetrahedron that has units at the tops of the element and in the middle of its edges; it is based on a quadratic approximation of movements inside the element (Figure 11b).

An oblique tetrahedron allows more accurate description of the geometry and deformation process of the research object; however, it has 10 internal units and contains 30 unknown values, which is almost three times the number of unknowns for an ordinary tetrahedron with four units and, accordingly, with 12 unknowns values sought for.

Therefore, a classical tetrahedron will be used for a qualitative study of VAT parameters, and an oblique one will be used for more accurate, final calculations. In SolidsWork, which is used to solve these problems, these are TETRA4 and TETRA10 finite elements, respectively [8].

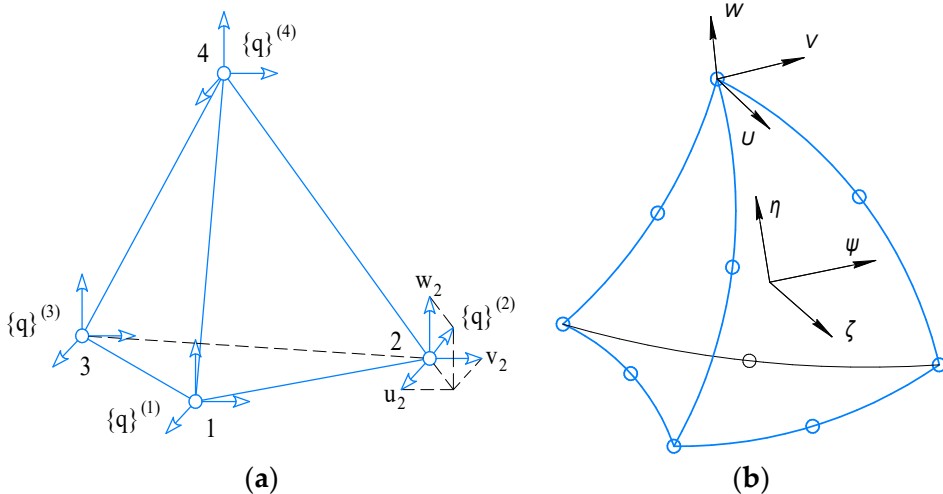

**Figure 11.** A finite element in the form of a tetrahedron: (**a**) TETRA4 and (**b**) TETRA10.

The influence of temperature fields leads to the emergence of additional temperature deformations [9–11]

$$\varepsilon_x^T = \alpha(T - T_0), \varepsilon_y^T = \alpha(T - T_0), \varepsilon_z^T = \alpha(T - T_0), \quad \gamma_{xy}^T = \gamma_{xz}^T = \gamma_{yz}^T = 0, \tag{4}$$

where $\alpha$ is the coefficient of linear temperature expansion of the material, $T = T(x, y, z)$—is the temperature distribution obtained from solving the thermal conductivity problem, and $T_0$ is the temperature at which there are no thermal stresses in the material.

The general deformation of the body consists of elastic deformations and temperature deformations [10]:

$$\begin{aligned}\varepsilon_x = \varepsilon_x^y + \varepsilon_x^T, \ldots \varepsilon_y = \varepsilon_y^y + \varepsilon_y^T, \ \varepsilon_z = \varepsilon_z^y + \varepsilon_z^T \\ \gamma_{xy} = \gamma_{xy}^y + \gamma_{xy}^T, \ \gamma_{yz} = \gamma_{yz}^y + \gamma_{yz}^T, \ \gamma_{xz} = \gamma_{xz}^y + \gamma_{xz}^T\end{aligned} \tag{5}$$

where $\varepsilon_x^y$, $\varepsilon_y^y$, $\varepsilon_z^y$, $\gamma_{xy}^y$, $\gamma_{yz}^y$, $\gamma_{xz}^y$ elastic deformations.

Taking into account temperature deformations, the relationship between stresses and deformations takes on the form:

$$\begin{aligned}\sigma_x &= \frac{E}{1+v}\left(\frac{v}{1-2v}\varepsilon + \varepsilon_x\right) - \frac{E\alpha}{1-v}(T - T_0), \\ \sigma_y &= \frac{E}{1+v}\left(\frac{v}{1-2v}\varepsilon + \varepsilon_y\right) - \frac{E\alpha}{1-v}(T - T_0), \\ \sigma_z &= \frac{E}{1+v}\left(\frac{v}{1-2v}\varepsilon + \varepsilon_z\right) - \frac{E\alpha}{1-v}(T - T_0), \\ \tau_{xy} &= G\gamma_{xy}, \tau_{yz} = G\gamma_{yz}, \quad \tau_{xz} = G\gamma_{xz}.\end{aligned} \tag{6}$$

Let us imagine (6) in matrix form:

$$\{\sigma\} = [B]\{\varepsilon\} - [T], \tag{7}$$

where $\{\sigma\} = \{\sigma_x\sigma_y\sigma_z\tau_{xz}\tau_{yz}\tau_{zy}\}^T$.

In Ref. [12] the methods for mesh decomposition have been submitted; however, this method cannot be fully applied due to the significant difference between fasteners and main structural elements. A significant contribution to mesh adaptation can be made by improved shape optimization based on mesh [13]. When constructing the mesh, we relied on the results given in Ref. [14]. It should be noted that, when using second order equations in numerical methods, the mesh was smoothed at the contact points of the elements. Taking into account the results of [14–18], the mesh was adapted, while at the attachment points and small elements, the mesh was thickened from the condition that, for conical surfaces, the stress during mesh discrediting should not differ by more than 0.1%.

In Figure 12 show basic grid for calculation.

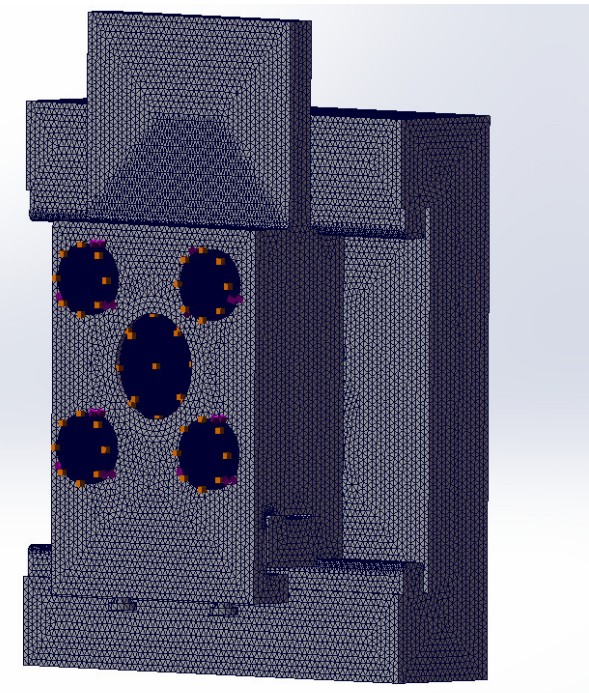

**Figure 12.** Calculation grid.

The stress–strain state of the assembled unit at compression is shown in Figure 13a, and when stretched is shown in Figure 13b.

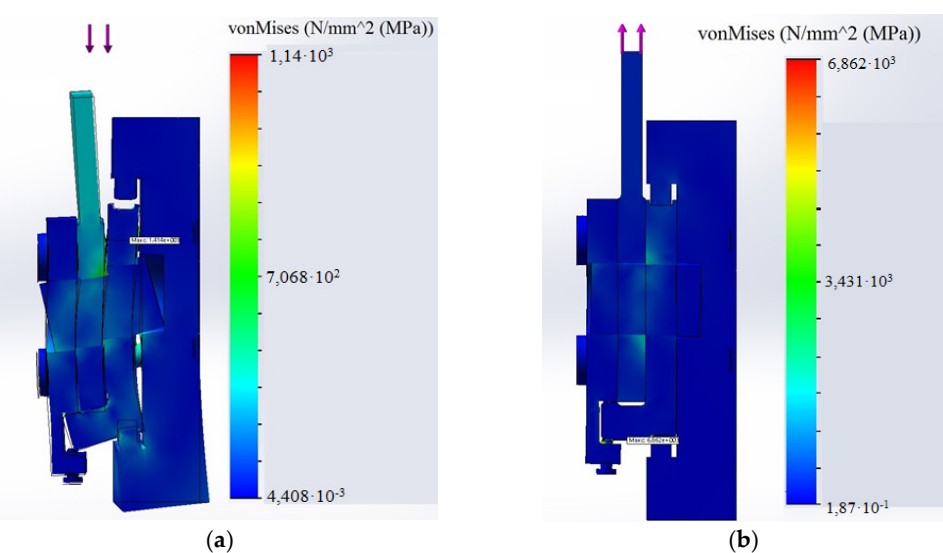

**Figure 13.** The Stress–Strain State of the Suspension: at Compression (**a**) and Stretching (**b**).

In Figures 14 and 15, the stress field on the spring surface and the diagram of stress changes along the curve, which is marked with numbers from 1 to 8 in the Figure, during compression is shown. As expected, there is a significant stress concentration near the holes.

The total stress values were analyzed for von Mises stresses. With that, the experience presented in Refs. [19–22] was taken into account.

In the process of correlation of the calculations obtained by the FEM, the values calculated using engineering methods were compared, as well as the actual stresses obtained by measuring with strain gauges.

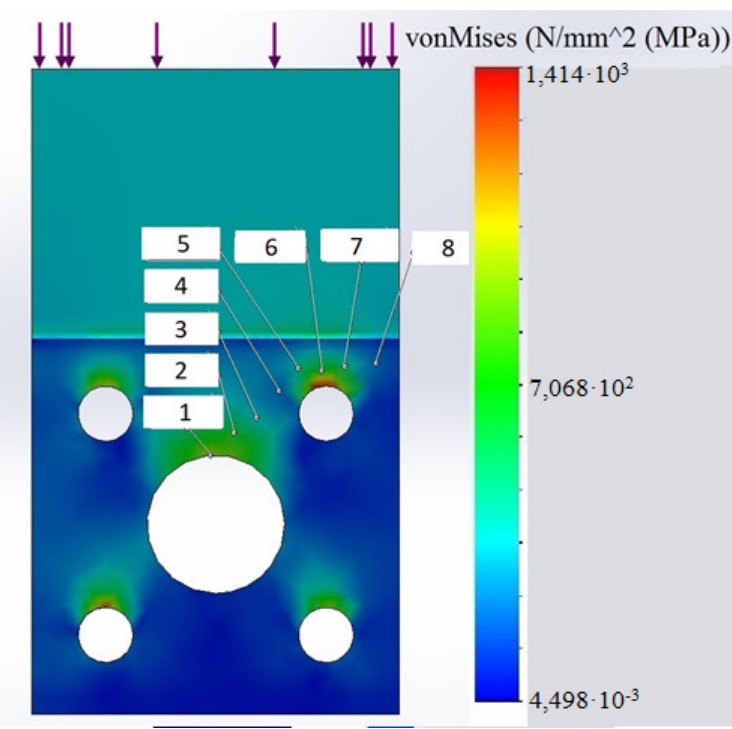

**Figure 14.** Stresses Field on the Spring Surface.

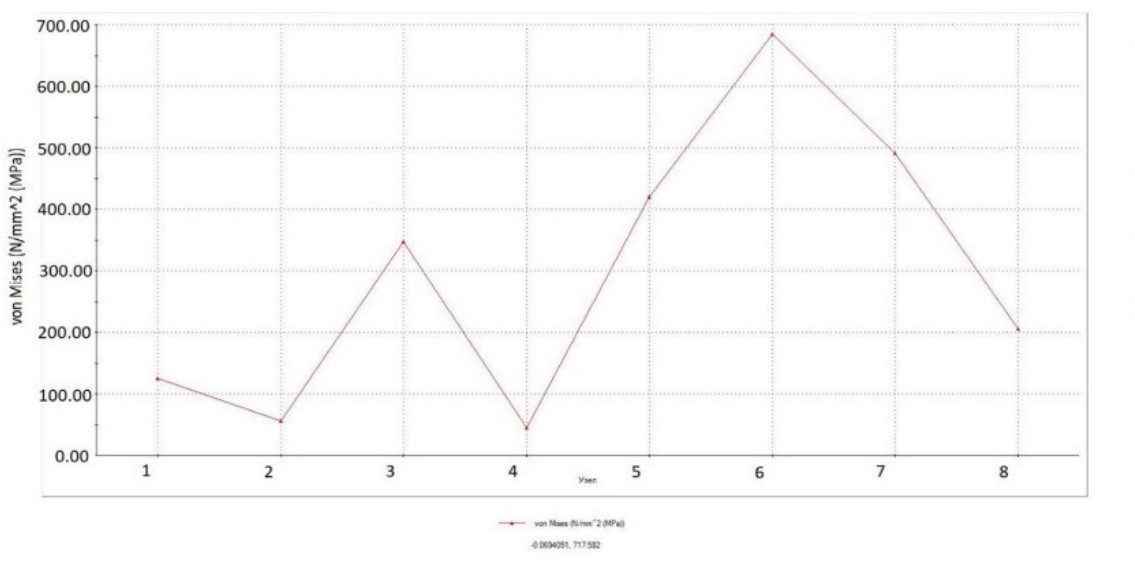

**Figure 15.** Stresses Change Diagram.

## 4. Discussion

Among the submitted scientific papers on the design of turbogenerators, a great number of works have been devoted to optimization of the design. Moreover, a significant amount of them are aimed at reducing the mass of the structure. In recent decades, new designs from Chinese and European schools have appeared, which have the most affordable prices with a moderate weight, but further optimization leads to a decrease in the projected resource. It is possible that in ten or twenty years we will face fatigue phenomena caused by vibration loads. As evidenced by the works of American and Canadian scientists, the new designs will have a reduced weight of the body parts, but the issue of increasing the reliability of the suspension of turbogenerators is widespread. The replacement of these units is not possible during operation, and if the springs fail, it will lead to the

destruction of the entire structure, including the turbines. Therefore, it is necessary to further determine the permissible limits of yield and quality of steel and declare them at the global worldwide level.

## 5. Conclusions

Table 1 shows the calculation data of the suspension assembly unit obtained by the engineering method and proposed method based on three-dimensional modeling. The stresses in the spring are caused only by compressive forces. It can be seen that the maximum deviation of calculation results according to the proposed method and according to the engineering method does not exceed 15%. This, on the one hand, confirms the reliability of the obtained results, and on the other hand, indicates the need to carry out final calculations on the strength of the suspension unit, using three-dimensional modeling to clarify the obtained stress values.

**Table 1.** Stress in the suspension of a generator rated 325 MW.

| Parameters | Calculation Method | |
| --- | --- | --- |
| | Engineering Calculation | 3D Calculation (Proposed Method) |
| maximum values of stresses in the spring, MPa | 40 | 44.6 |
| stress in pins Ø60: from crumpling (spring-pin), MPa | 85.6 | 84.4 |
| crumpling stress between pin Ø60 and support (strap), MPa | 22.7 | 25 |

**Author Contributions:** Writing—review and editing O.T., D.K., I.K., V.S., M.A., I.T., H.D. and V.N. All authors have read and agreed to the published version of the manuscript.

**Funding:** This research received no external funding.

**Conflicts of Interest:** The authors declare no conflict of interest.

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
