# Peer review of "Modeling of the Stress–Strain of the Suspensions of the Stators of High-Power Turbogenerators"

_computation, doi:10.3390/computation10110191_

Round 1

Reviewer 1 Report

The article analysed the crucial scientific problem regarding turbogenerators' operation life and effectiveness. This work presents the mathematical description and FEM analysis. The presented issue is very interesting and needs to be analysed. The problem description is proper and adequate. However, the modelling and simulation analysis is insufficient. The authors should present the analysis more precisely and widen the FEM analysis and discussion. From a scientific point of view, the current description of modelling analysis is deficient.

The bibliography linked to the analysed scientific problem is relevant and sufficient. Symbols (constants, variables) and functions are appropriately described and explained.

The paper contains the following detailed faults:

 The title of the 2nd chapter should be cut short.

 In Fig. 4, the dimensional parameters of the assembly should be presented.

   For a clear paper description, The 2nd chapter should be divided into two parts to separate the theory analysis and FEM analysis parts.;

All in all, the article requires revision before publication, such as enhancement of FEM analysis and discussion part.

Author Response

Dear Reviewer,

Thank you for the work done in reviewing our article.

Reviewer 2 Report

In this article, the authors showed a case of the existing types of stator fastening design of turbo generators, and the main causes of the stressed state of the stator suspensions are considered. A detailed calculation of the complex stressed state of the turbogenerator stator suspension was carried out, taking into consideration the unevenness of the heat distribution along the horizontal axis of the unit for a number of electrical sheet steels.

- Main contribution to the field:

The article is within the scope of the magazine.

- Quality of experiments, simulations, etc.:

Measurements were carried out with adequate quality.

- Importance of findings:

This solution is interesting. The results of the experiments and simulations show compatibility with the research carried out.

- Clarity of presentation:

The article was written with a coherent cause-effect sequence. Despite the poor form of the article. Individual frame segments of the manuscript have been preserved and discussed.

- Length of the manuscript and whether it should be reduced. :

The length of the manuscript is appropriate.

- The authors cite fairly outdated literature in references. There the article is only five publications published in the article - this undermines the innovation of the solution presented - please correct this.

- This article could be published, provided that the quality and format are improved very deeply.  I believe that the quality of the drawings should be improved and the fonts of the parameters and descriptions in the drawings should be unified to make them legible. (In this form it looks unsightly).

- Please improve the tables according to the template of MDPI magazine

- Please add and cite current literature and expand the "Introduction" section with new publications to highlight the "novelty" of the article.

Author Response

(The authors gave the same response as above.)

Round 2

Reviewer 1 Report

Dear Authors, thank you for the article correction and his supplementation in the initial chapters. Now, the first part of the paper is better and clear. However, the scientific analysis is still insufficient, so the analysis should be widened and precisely described. It concerns the FEM analysis and the result's discussion. It still needs to be supplemented in terms of study and effect analysis (i.e. advantages and disadvantages of the modelling analysis, the predicted experiment effects, and the influence on further research, etc.) Summing up, the paper is unsatisfactory from a scientific point of view. 

Author Response

(The authors gave the same response as above.)

Round 3

Reviewer 1 Report

The corrected paper is adequate to be published in a scientific monograph.

All in all, I recommend the article for publication in its present form.